# Influence of Variable Chloride/Sulfur Doses as Part of Potassium Fertilization on Nitrogen Use Efficiency by Coffee

**DOI:** 10.3390/plants12102033

**Published:** 2023-05-19

**Authors:** Victor Hugo Ramirez-Builes, Jürgen Küsters, Ellen Thiele, Luis Alfredo Leal-Varon, Jorge Arteta-Vizcaino

**Affiliations:** 1Center for Plant Nutrition and Environmental Research Hanninghof, Yara International, 48249 Dülmen, Germany; 2Yara Colombia, Zona Industrial Mamonal, Cartagena 130009, Colombia; 3Department of Plant Science, National University, Palmira 763533, Colombia

**Keywords:** coffee, chloride (Cl^−^), sulfur (S), nutrients, NUE

## Abstract

Chloride (Cl^−^) is applied in coffee at rates as a “macronutrient” in the form of muriate of potash (MOP). Potassium (K^+^) is one of the most demanded nutrients by the coffee plant, and MOP is one of the most used fertilizers in coffee production. No scientific evidence shows how Cl^−^ applied with MOP influences coffee growth, nutrient uptake, and nitrogen use efficiency (NUE). In order to address these questions, a greenhouse trial over two years and a field trial over four years were conducted. The trials were designed to test the influence of variable Cl^−^/S ratios on biomass accumulation, nutrient uptake, and NUE. A significant effect of the Cl^−^ rates on growth was observed under greenhouse conditions but a non-significant effect on yield under field conditions. Cl^−^ and S significantly influenced the NUE in coffee. The results allow us to conclude that Cl^−^ rates need to be balanced with S rates, and that Cl^−^ applied at macronutrient rates can improve the NUE in coffee between 7 and 21% in greenhouse conditions and between 9% and 14% in field conditions, as long as the rates do not exceed 180 mg L^−1^ Cl^−^ and 80 mg·L^−1^ S in the greenhouse and 150 kg·ha^−1^·year^−1^ Cl^−^ and 50 kg ha^−1^·year^−1^ S in the field. With the aim to improve the NUE in coffee, the Cl^−^ content in leaves in coffee should be lower than 0.33% of dry matter, and in soil lower than 30 mg·L^−1^. In practical terms, coffee farmers need to balance K-based fertilizers to avoid the excessive Cl^−^ applications that reduce the nutrient use efficiency, especially the NUE.

## 1. Introduction

In crop production, chloride (Cl^−^) is still being seen as an undesired anion rather than a plant nutrient, mainly because of its toxicity effects resulting from excessive Cl^−^ accumulation in sensitive plant organs under salt stress [1,2]. There is also a widespread belief that Cl^−^ and nitrate (NO_3_^−^) are antagonistic ions competing for plant uptake [3]. However, since 1954, chloride has been considered as an essential micronutrient, when Broyer et al. [4] demonstrated a direct influence of Cl^−^ on the growth of roots, leaves, stems, and petioles in tomato plants. Chloride is involved in important physiological processes, including stomatal regulation, water splitting or Hill reactions in photosystem II. Cl^−^ also has an osmoregulatory function in the vacuole, participating in phloem loading and unloading of sugars and stimulating membrane-bound proton-pumping ATPases and PPiases [4,5]. Other important functions are related to balancing the electrical charge of essential cations, such as K^+^ and H^+^, which both play a main role in stabilizing the cell membranes’ electrical potential and which regulate pH gradients [6].

Being a non-assimilating highly mobile anion, Cl^−^ is the preferred molecule to balance the electrical charge of important cations such as K^+^, Ca^+2^, and protons (H^+^), helping in the stabilization of the electrical potential of cell membranes and the regulation of the pH gradients and electrical excitability [2]. Cl^−^ is directly involved in cell division and, subsequently, leaf area formation [2,7]. Most recently, a positive influence of Cl^−^ on the vegetative yield per unit of nitrogen available to the crop (NUE) in different crop species has been documented [8].

Chloride occurs predominantly as Cl^−^ in the soil, does not form complexes readily, and tends to be repelled by the negatively charged sites on layer silicates. The concentration of Cl^−^ in bulk solutions is greater than in the diffuse layers surrounding the soil particles [1]. Cl^−^ redistribution in the soil profile is likely to be dominated by mass flow and convention processes associated with water fluxes [2]. The surface complexation of Cl^−^ by soil particles is relevant in variable-charge soils, where two types of soil complexes are distinguished: outer sphere and inner sphere complexes. In outer sphere complexes, the ion is separated from the particle surfaces by water molecules, and bonding depends solely on the presence of opposite charge on the particle surfaces. Cl^−^, NO_3_^−^ and many cations (K^+^, Mg^+2^, Na^+^) form only outer-sphere complexes [9]. Permanent positive surface charge occurs, but outer-sphere anion complexation is important mainly in variable charged soil with a pH below the point zero charge [9].

N fertilizer application in coffee is relatively high, with rates of 200 to 500 kg·ha^−1^·year^−1^ [10,11,12,13,14]. The efficiency of mineral N fertilizer or NUE depends on the N rate applied and the crop N uptake at a given yield level. Bruno et al. [12] reported an N recovery for the whole coffee plant at harvest time of 66% when the N rate was 200 kg N·ha^−1^ and only 37% when a high N rate of 800 kg N·ha^−1^ was used. Cannavo et al. [15] reported that 60% of the N applied to the coffee crop remained in the top 60 cm of the soil, meaning that not more than 40% of the N applied was used by the coffee crop. In nursery coffee plants, Salamanca et al. [16] found that fertilizer N supplied approximately 20–29% of the total plant N after 4 months and an NUE of lower than 10%. The above reports demonstrate that the NUE in coffee needs to be improved and that integrated strategies must be sought from plant breeding to agronomic management. Within the agronomic management strategies, the appropriate use of sources and doses of fertilizers in conjunction with the times of application are part of the pool of alternatives that need to be considered with the aim to improve the NUE in coffee.

The most often used potassium source by coffee farmers is the muriate of potassium (MOP), also known as potassium chloride (60% K_2_O, 46% Cl), because of its high K concentration, high solubility, and relatively low costs compared to other K sources such as potassium sulfate (SOP) and potassium nitrate (PN). The potassium demand of the coffee crop is similar to the nitrogen demand, and in some phenological stages, it is even higher. The mean application of K by the coffee production systems changes from region to region, with K fertilization rates between 100 and 400 kg K_2_O ha^−1^·year^−1^ depending on the expected yield and K content in the soil [14,17,18,19]. In Colombia, for example, the economic optimum coffee yield is achieved with a mean rate of 273 kg K_2_O ha^−1^·year^−1^ in productive coffee plantations [13]. High K_2_O rates applied with MOP result in high inputs of Cl^−^, reaching up to 300 kg Cl^−^·ha^−1^·year^−1^.

In weathered tropical soils, the natural content of Cl^−^ tends to be low. Cl^−^ as an anion is not adsorbed by the soil particles and hence is very mobile in the soil. It can be easily lost by leaching under freely drained conditions [2,20].

Laboratories in coffee regions usually have well-standardized analytical methods to quantify Cl^−^ content in the soil and tissues; however, despite its role as a plant nutrient, this element is not included as a key nutrient in any of the nutritional programs in the coffee regions. Only in fertigation is special emphasis made that Cl^−^ should not exceed 350 mg·L^−1^ to avoid a reduction in the production capacity of the trees [14].

In rainfed conditions, Cl^−^ is the third most applied nutrient in coffee when MOP is used to supply the K^+^ needs of the coffee crop. Excess supply and uptake of Cl^−^ in plants is a serious problem, mainly in crops that grow on salt-affected soils [1,20]. Recently, Santos et al. [21] demonstrated the negative effect of chloride applied via fertilizers in coffee with respect to productivity and coffee cup quality in the Minas Gerais state of Brazil, but no references are available regarding the effects of high Cl^−^ supplies on nitrogen use efficiency (NUE) in coffee growing under rainfed conditions. Hence, this research aimed to evaluate the influence of Cl^−^ applied at rates as a macronutrient for coffee regarding growth, productivity, nutrient uptake, and NUE in coffee.

## 2. Results

### 2.1. Influence of Cl and S Rates on Biomass Accumulation, Nutrient Uptake, and NUE under Greenhouse Conditions

The dry biomass was significantly affected by the Cl/S rates (*p* value < 0.001). The treatments without S and with a Cl^−^ rate of 300 mg·L^−1^ showed 2,2 times less biomass (37.9 g·plant^−1^) with respect to the treatments without Cl and 200 mg·L^−1^ of S that accumulated at 82.1 g·plant^−1^ (Table 1). This reduction can be basically attributed to the strong S deficit observed in this treatment, where this low dry biomass accumulation reduces the total N uptake, NU_T_E, and NUE, clearly indicating the importance of S on coffee growth and NUE. Otherwise, the treatment without Cl^−^ and the highest S rate (200 mg·L^−1^) showed the highest biomass accumulation and NU_T_E but not the highest NUE. The intermediate treatment with 180 mg·L^−1^ of Cl^−^ and 80 mg·L^−1^ of S showed the highest total N uptake (1.2811 mg. plant^−1^) and NUE (0.89 g shoot DW·g^−1^ of N applied) but significantly lower NU_T_E (0.038 g DW·mg^−1^ N).

### 2.2. Cl/S Ratios Influence Nutrient Concentration and Uptake by Coffee Beans under Field Conditions

In the field conditions where the present study was carried out, it was possible to observe the occurrence of two dry periods during the year, represented by a soil moisture index (SMI) lower than 1.0, during the months of July to September and from December to March. The greatest flowering occurred in the months of February to March associated with a greater water deficit, with SMI values between 0.7 and 0.5, indicating that between 70% and 50% of the pore space of the soil is occupied by water (Figure 1).

In field conditions on productive coffee plants, the nutrient concentration of the coffee cherries after flowering was reduced in the way that the fresh cherries develop and gain weight. Nitrogen (N) concentrations move from a mean value of 3.6% 30 days after flowering to 1.8% 240 days after flowering or harvest time. At harvest time, the N concentration was significantly different between treatments (Table 2), where the treatment with the higher Cl/S ratio (201 kg Cl/33 kg S) showed the lowest N concentration with a mean value of 1.69% compared with the treatment with 150 kg Cl/50 kg S, which showed a mean N concentration of 1.89% (Figure 2A).

The potassium (K^+^) concentration of the coffee cherries moves from average values of 2.7% 30 days after flowering to 1.9% 240 days after flowering. Significant differences between treatments were observed in potassium (K^+^) at 60 and 150 days after flowering (Table 2). Sixty days after flowering, the treatment with 201 kg Cl/33S showed a significantly lower K^+^ concentration of 2.27% than the treatment with 150 kg Cl/50, which shows 2.5% of K^+^ on the cherries. Furthermore, 150 days after flowering, the treatment without Cl^−^ and high S (0 kg Cl/125 kg S) showed the lowest and most significant different K^+^ concentration for the cherries at 1.95% compared with the treatment with a higher Cl/S ratio of 201 kg Cl/33 kg S with 2.14% of K^+^ concentration for the cherries (Figure 2B). At harvest time, no significant differences were observed in K concentration for the coffee cherries, but the treatment with a higher Cl/S ratio (201 kg Cl/33 kg S) showed the lowest K^+^ concentration for the cherries with 1.82% concerning the other treatments, where the K^+^ concentration tended to be higher.

The calcium (Ca^+2^) concentration of the coffee cherries moves from a mean value of 1.6% 30 days after flowering to 0.36% 240 days after flowering, without any significant differences between treatments during the coffee cherry’s development period (Figure 2C).

The magnesium (Mg^+2^) concentration in the coffee cherries moves from a mean value of 0.61% 30 days after flowering to 0.12% 240 days after flowering (Figure 2D), with significant differences between treatments 60 days after flowering (Table 2), where the treatment with a higher Cl/S ratio showed the lowest Mg^+2^ concentration of 0.25% with respect to the treatment with 150 kg Cl/50 kg S, which registered a concentration of Mg^+2^ of 0.30% at the development stage. At harvest time, no significant differences were observed in the Mg^+2^ concentration in the coffee cherries.

The sulfur (S) concentration of the coffee cherries moves from a mean value of 0.32% 30 days after flowering to 0.15% 240 days after flowering (Figure 2E), with significant differences between treatments observed 210 days after flowering (Table 2). The treatment with a medium Cl/S ratio (101 kg Cl and 73 kg S) showed a lower S concentration of 0.12%. No significant differences in the S concentration for the cherries were observed at harvest time (240 days after flowering).

The Cl^−^ was the only nutrient that did not reduce the concentration during the cherries’ development, as was described with the other nutrients previously. At 30 days after flowering, the Cl^−^ concentration for the coffee cherries showed a mean value of 0.16%, while 120 days after flowering, the Cl^−^ concentration was six times higher, reaching a mean value of 0.92%, with a subsequent reduction 240 days after flowering, reaching a mean value of 0.028% (Figure 2F).

The Cl/S ratios significantly influence the total N and Cl^−^ uptake per ton of green coffee beans. For N, the treatment with 150 kg Cl/50 kg S showed a mean N uptake of 39.13 kg t^−1^ of green coffee beans, 14% more N uptake per ton of green coffee than the treatment with 201 kg Cl/33 kg S, which showed a mean N uptake of 34.2 kg N·t^−1^, followed by the treatments with 101 kg Cl/73S with 37.5 kg N·t^−1^ and the treatment with 53 kg Cl/97 kg S with 37.9 kg N·t^−1^, representing 9% and 11% more N uptake per ton of green coffee bean, respectively (Figure 3A).

The Cl^−^ uptake per ton of green coffee bean increased proportionately with the increase in the Cl/S ratio, treatments without and with low Cl^−^ presented lower and significantly different Cl^−^ uptake per ton of green coffee, with 0.497 kg Cl·t^−1^ of green coffee bean for the treatment with 0 kg Cl/123 kg S, and 0.336 kg Cl·t^−1^ of green coffee beans for the treatment with 53 kg Cl/97 kg S. This represents 65% and 144% less Cl^−^ uptake with respect to the treatment with the higher Cl^−^ rate (201 kg Cl/53S) where the mean uptake was 0.82 kg Cl·t^−1^ of green coffee beans (Figure 3B).

### 2.3. Cl/S Ratios Influence Nitrogen Use Efficiency for Coffee Beans at the Field Level

The NUE in the field trial, described as the ratio of N uptake by the green coffee beans with respect to the mineral N applied, was significantly influenced by the age of the plantation and by yield (Figure 4). For instance, the NUE for the first harvest after stem pruning was lower than 0.25 kg N uptake per kg of N applied, while in the second harvest, it increased until a mean value of 0.45 kg N uptake per kg of N applied reached the highest level during the third harvest year, with a mean value of 0.75 kg N uptakekg N applied^−1^.

The NUE for the productive plants shows a polynomial correlation with respect to Cl^−^ rates (Figure 4) during the second and third harvest years. A significant polynomial correlation between the Cl^−^ rates and the NUE was observed (R^2^ = 0.54 for the second harvest and R^2^ = 0.97 for the third harvest). The polynomial function reaches an optimal NUE with Cl^−^ rates of 73, 113, and 110 kg·ha^−1^ during the first, second, and third harvests, respectively (Figure 4).

The increase in NUE over the years is directly related to the increase in yield. During the first and second year after pruning (first harvest), the Cl^−^ rates were low due to the low yield potential of the crop and the low K^+^ demand. After the second year after pruning, yield potential increases, and crop nutrient demands, including K^+^, also increase. 

The Cl/S rates do not have a significant effect on coffee yield (Table 3). The treatment with a higher Cl/S ratio (201 kg Cl/33 kg S) for 4 years and three harvests showed only a 3% lower yield compared with the treatments with lower Cl/S ratios such as 145 kg Cl/50 kg S or 53 kg Cl/97 kg S, but those treatments showed an improvement in NUE with a mean value of 14% and 16%, respectively.

On average, the NUE ranges between 0.44 and 0.51 kg of green coffee per kg of N applied in the treatments with a higher Cl/S ratio (201 kg Cl/33 kg S) and a lower Cl/S ratio (53 kg Cl/97 kg S), respectively (Table 3).

### 2.4. Cl/S Ratios Influence Cl and S Distribution in the Soil Profile

Significant differences in the Cl^−^ concentration in the soil were observed between treatments at soil depths of 10 and 20 cm. The treatment with a higher Cl/S ratio (201 kg Cl/33 kg S) showed a higher concentration with a mean value of 95.8 mg·L^−1^ of Cl^−^ at 10 cm depth and 66.4 mg.L^−1^ at 20 cm depth, which was significantly different from the other treatments were the mean Cl^−^ concentrations at those depths ranged between 27 mg·L^−1^ at 10 cm depth and 18 mg·L^−1^ at 20 cm depth in the treatment with 101 kg Cl/73 kg S, reaching the lowest concentration in the treatment with 0 kg Cl/125 kg S with a mean Cl^−^ concentration of 9.4 mg·L^−1^ at 10 cm depth and 10.18 mg·L^−1^ at 20 cm depth (Figure 5).

These results clearly show a reduction in Cl^−^ concentration in the soil for the treatments with less Cl^−^ applied within a specific soil depth. At 50 cm depth, no significant differences in the Cl^−^ concentration were observed between treatments (Figure 5).

In the case of S, no significant differences were observed between treatments, but there were significant differences between soil depths. In the first 20 cm, the treatment with the higher Cl/S ratio (201 kg Cl/33 kg S) had a mean S content of 15.6 mg·kg^−1^, while the treatment without Cl^−^ and the higher S rate (0 kg Cl/125 kg S), showing a mean value of 23.4 mg·kg^−1^, but at a 50 cm depth, the treatment without Cl^−^ and the higher S rates (0 kg Cl/125 kg S) increase the S concentration significantly until 71.8 mg·kg^−1^, while the treatments with lower S rates and medium Cl^−^ rates (150 kg Cl/50 kg S) showed a lower S concentration of 11.92 mg·kg^−1^ at 50 cm depth.

## 3. Discussion

### 3.1. Cl^−^ Concentration on Tissues and Influence on Growth and Productivity

Cl^−^ does not appear as a typical micronutrient since the actual Cl^−^ concentration in plants is in the range of 0.2% to 2% of dry matter [2,6]. In most plant species, the Cl^−^ requirements for optimal plant growth, however, are in the range of 0.02% to 0.04% of dry matter [3,5], which corresponds to the content of a micronutrient, and the Cl^−^ content available in nature is sufficient to fulfill these requirements [3]. The critical tissue Cl^−^ concentration for toxicity is about 0.4% to 0.7% for Cl^−^-sensitive and 1.5% to 5.0% for Cl^−^-tolerant plant species [1]. In this research, the Cl^−^ concentration on the leaves changes according to the Cl^−^ rate from 0.03% in the treatment without Cl^−^ to 2.98% of dry matter for the treatment with 300 mg L^−1^ Cl^−^ without S. These increases in Cl^−^ concentration in the greenhouse trial had a significant effect on dry biomass accumulation, with a significant reduction on increasing Cl^−^ rates (Table 1). The significant reduction in biomass accumulation in the greenhouse trial with leaf Cl^−^ content above 0.33% places coffee in the group of Cl^−^-sensitive glycophytic plants.

According to Chen et al. [2], when Cl^−^ levels are high enough to be toxic, cation absorption, such as for K^+^, decreases because of the disordered cell metabolism. In the greenhouse trial, we see this tendency with the significant reduction in the dry biomass accumulation when the Cl^−^ rates increase from 0 to 60 mg L^−1^ (Table 1), and in the field conditions, a significant reduction in the K^+^ concentration on the coffee cherries was observed 60 days after flowering but with no significant effect on K^+^ uptake at harvest time. (Table 2, Figure 2).

The Cl^−^ content in grains, fruits, and seeds is very low and is hardly affected by the Cl^−^ concentration of the soil solution [23]. In the case of the field trial, we observe strong changes in the Cl^−^ concentration in the coffee cherries during the development process that was directly linked with the fertilizer application time and the soil moisture changes during the year (Figure 1 and Figure 2). In the work of Silva et al. [24], who compared the influence of K sources on coffee productivity and quality, it is possible to observe 30% less Cl^−^ concentration in the coffee beans on the treatments with SOP compared with the MOP, with significant impact on the coffee quality parameters such as total sugar content and polyphenol oxidase activity.

Cl^−^ application stimulates plant growth when it is supplied at macronutrient levels [8]. Root Cl^−^ uptake and long-distance transport require considerable use of metabolic energy, clearly indicating that shoot Cl^−^ accumulation on macronutrient levels responds to specific biological adaptation [3]. The increase in biomass production induced by the higher rates of Cl^−^ application as a macronutrient is associated with the stimulation of higher turgor, cell size, and shoot expansion [3,6]. Studies in soils with high to very high extractable K^+^ levels on wheat and alfalfa in Argentina showed a positive influence of Cl^−^ fertilization using MOP and ammonium chloride as a fertilizer source [25]. In both cases, the authors report 50% yield increases independently of the fertilizer sources with Cl^−^ rates between 23 and 56 kg ha^−1^. In anion crops, Cl^−^ supplying higher levels up to 500 mg kg^−1^ on the nutrient solution shows that Cl^−^ on average is the fourth most utilized essential element, superseded only by N, K, and P [26]. In durum wheat under field conditions, it has been proven that a number of physiological disorders impairing growth and yield are especially due to Cl^−^ deficiency [27]. Plants such as kiwi fruit and palm trees have higher Cl^−^ requirements, which cannot be alleviated through NO_3_^−^ addition, and the reasons for such high Cl^−^ demand is still unknown [3,28,29].

The concept of beneficial elements became popular around the early 1980s to include elements that stimulate plant growth or health but have not been shown thus far to meet the strict essentiality criteria [30], or not essential in certain plant species, or under specific conditions [31]. In this investigation, in both coffee trials, no stimulating effect of the application of Cl^−^ on growth and yield was observed (Table 1 and Table 3), nor an increase in biomass accumulation in the greenhouse with Cl^−^ rates higher than 60 mg·L^−1^. At field conditions, in Brazil, Santos et al. [21] reported a yield reduction with an application of 100% of the K^+^ as a MOP, which represented an average Cl^−^ dose of 230 kg Cl·ha^−1^·year^−1^.

A negative effect could be observed in some crops when the rates of Cl^−^ increase to 200–400 mg·kg^−1^. For most crops, the negative effect was obvious when the applied amount increased to 400–500 mg·kg^−1^, and the yield of most crops decreased rapidly when the applied Cl^−^ exceeded 800 mg·kg^−1^ [2].

### 3.2. Cl^−^ Influence on Nutrients Uptake and NUE

Cl^−^ plays a quantitatively important role in ion balance when Cl^−^ is abundant, but other anions (nitrate, malate) can fulfill this role when Cl^−^ supply is reduced. Competitive effects in uptake between Cl^−^ and N-NO_3_^−^ and Cl^−^ and SO_4_^2−^ were documented by De Wit et al. [32], who recently reported that Cl^−^ somewhat affects the uptakes and utilities of N, P, K, Ca, Mn, Si, S, Zn, Mg, Fe, and Cu in potatoes, with the most extreme competitive effects of N-NO_3_^−^ in crops such as rice, corn, soybean, cabbage, tomato, strawberry, melon and lettuce, peanut, barley, citrus, and spring wheat [2,23].

Cl^−^ in excess can strongly reduce the NUE specifically interfering with its uptake, transport, and loading into the root xylem, since it uses the same anion channels used by NO_3_^−^ [33,34]. NO_3_^−^ and Cl^−^ are the most abundant inorganic anions in plants and share similar physical properties and transport mechanisms, which is the origin of the strong dynamic interactions between these two monovalent anions and which frequently explains why the higher accumulation of Cl^−^ leads to lower NO_3_^−^ content in plants [3]. This antagonistic interaction between Cl^−^ and NO_3_^−^ has been reported by several authors [20,23,35] and is one of the reasons why Cl^−^ is considered a detrimental nutrient in agriculture.

Colmenero-Flores et al. [3] have shown that prolongated exposure to a nutrient solution containing Cl^−^ at a concentration of 4–5 mM (140–180 ppm) may cause a gradual non-toxic accumulation of Cl^−^ at values ranging between 2.5% and 5.0% DW (macronutrient levels), without any interferences on plant growth and stress symptoms. According to Carillo and Rouphael [34], when Cl^−^ is in excess, it is passively transported into the cortical cell and the xylem by anion channels, such as the NO_3_^−^ transporter NPF7.3 and S-type anion heteromeric channel SLAH1/SLAH3, where high Cl^−^ concentration at the leaf level is less controlled and more dangerous than that of sodium due to the lower capacity of the leaf blade to exclude Cl^−^ and its limited basipetal phloem transport toward the roots.

When Cl^−^ is accumulated in high concentration in the leaf tissues, it initially decreases the apoplast osmotic potential interfering with the cellular water relations [35]. Thereafter, it diffuses into the symplast by using anion (e.g., nitrate and phosphate) uptake symporters competing with these beneficial nutrients for uptake within the cell [35].

In coffee growing under greenhouse conditions, the highest doses of Cl^−^ without S strongly reduce N uptake, the utilization efficiency—NU_T_E, and NUE. Treatments without Cl^−^ showed the highest NU_T_E but not the highest N uptake and NUE. Treatments with low to medium Cl^−^ rates (60 to 180 mg L^−1^) significantly increased NUE without significant changes in NU_T_E (Table 3).

At the field level, the results showed significant differences in N uptake by the coffee cherries (Figure 3) according to the Cl^−^ and S rates, the treatment with high Cl^−^ (201 kg Cl/33 kg S) showed significantly lower N uptake than the treatment without Cl^−^ (0 kg Cl/125 kg S), but the higher and significantly different nitrogen uptake was achieved in the treatment with medium Cl^−^ rate (between 100–150 kg Cl/73–50 kg S), in which a similar tendency was observed for NUE (Figure 4).

Nitrate uptake and allocation are key factors regulating NUE [36], given the close interaction between Cl^−^ and NO_3_^−^, and it is expected that Cl^−^ can significantly influence the NUE, as can be seen in both coffee trials, where the NUE was reduced in the higher Cl^−^ rates, but the question is, why in medium or low Cl^−^ rates was the NUE improved? Can we say that Cl^−^ at a certain point can improve the NUE? The potential reason could be related to the net NO_3_^−^ uptake dynamic, which results from the differences between NO_3_^−^ influx mediated by active transport and its passive efflux through anion channels [3]. Root anion efflux to the rhizosphere might be important in regulating H^+^-ATPase activity, maintaining the H^+^ charge balance [37], or regulating plant cell growth [3]. The release of Cl^−^ from root cells through anion channels, replacing NO_3_^−^ efflux, could be an important mechanism for preventing N loss [38], which is expected to improve NUE [3].

Rosales et al. [8] suggest that Cl^−^ nutrition reduces NO_3_^−^ sequestration in plant leaf tissues (e.g., vacuolar compartmentalization), making this valuable N source available for assimilation and biosynthesis of organic N. Cl^−^ can improve the NUE, despite significantly reducing foliar NO_3_^−^ storage, which represents a radical change in the perception of the Cl^−^ and NO_3_^−^ antagonist. The most likely scenario is that when NO_3_^−^ is available, the active transport mechanism that is frequently more selective for NO_3_^−^ than for Cl^−^ prioritizes NO_3_^−^ influx by inhibiting Cl^−^ uptake. When little NO_3_^−^ is available, Cl^−^ influx is less inhibited, thus increasing root uptake and intracellular Cl^−^ concentration, which is expected to replace NO_3_^−^ in serving an osmotic function, allowing for more efficient use of the available N [2].

The stimulatory effect of Cl^−^ on the asparagine synthetase activity has also been suggested as another mechanism of interaction between Cl^−^ and NUE [3]. Chloride increases the affinity of asparagine synthetase for glutamate, its substrate [39]. Asparagine is a major compound in the long-distance transport of soluble N in many plant species [3,5], which explains the higher accumulation of N in coffee cherries with medium Cl^−^ rates and the higher NUE compared with the treatments without Cl^−^.

Rosales et al. [8] reported that Cl^−^ significantly increases the NUE in different crops such as tobacco, olive, mandarin, lettuce, spinach, and chard when accumulated at the macronutrient level. Finely modulating the Cl^−^ dose for decreasing the NO_3_^−^accumulation in leaves or improving its uptake and assimilation without decreasing the growth and productivity of the plants is necessary [34].

In the case of coffee at the field level, this fine modulation of Cl^−^ and S rates and ratios shows potential benefits to improving the NUE, as is indicated by these results, creating a fine balance for the Cl/S rates with mean values on a productive coffee plantation between 100 and 150 kg Cl^−^ ha^−1^. year^−1^ and 73–50 kg S·ha^−1^·year^−1^. We can hypothesize in this study that Cl^−^ improved NUE when applied at medium doses at the “macronutrient” level, through a reduction in N compartmentalization and improved transport through stimulation of asparagine synthetase, represented in higher N uptake by coffee cherries without a significant increase in yield in field conditions and biomass in greenhouse conditions (Figure 4, Table 1). However, such macronutrient rates of Cl^−^ did not improve NO_3_^−^ assimilation represented by the NU_T_E (Table 1), similarly reported in Cl^−^ excluding species such as olive and citrus rootstock Cleopatra mandarin plants [8].

In the field trial over four years, a significant yield reduction was not observed, likely related to the high soil moisture during the study time, with long periods of water excess between March and July and between October and December over the years (Figure 1), allowing for Cl^−^ movement into the soil profile. Although the Cl^−^ contents in the soil in the first 20 cm of depth were significantly higher in the treatment with 201 kg Cl and 33 kg S, the electrical conductivity (EC) was lower than 1.2 dS·m^−1^ in the soil’s first 10 cm and less than 0.5 dS·m^−1^ at 30 and 50 cm depths.

The discussion on the effect of Cl^−^ on the quality of coffee is still open. Silva et al. [24] demonstrated more than ten years ago a reduction in quality parameters with the application of MOP in coffee, and recently, Santos et al. [21] demonstrated a significant reduction in the quality assessment of coffee in treatments that had low and high proportions of Cl^−^ in mineral K fertilization. However, an interesting note in the study by Santos et al. [21] is that the treatment without Cl^−^ was not the one that presented the highest yields in coffee, while the treatment with 25 to 50% of K^+^ as MOP and 75% to 50% as SOP (57 to 115 kg Cl·ha^−1^·year^−1^) presented the highest yield, probably related to the effect of Cl^−^ on the NUE.

### 3.3. Cl^−^ on the Soil

The Cl^−^ content of the soil is not an intrinsic property of the soil but rather a result of soil management [23]. As Cl^−^ can move freely with soil water, soil Cl^−^ levels can be highly variable and can increase or decrease from year to year, depending on the water table and the location in the landscape [1,40]. Soils considered low in Cl^−^ are below 2 mg·kg^−1^. Regarding wheat in several soils of the United States, Fixen et al. [41] reported that higher Cl^−^ levels in the soil of 43.5 kg·ha^−1^ (0–0.6 m) were adequate for near-maximum wheat yield; in Argentina, Diaz-Zorita et al. [25] reported that the Cl^−^ levels in the soil higher than 13.2 mg·Kg^−1^ (0.0 to 0.2 m) were adequate for maximum grain yield.

In the present research on coffee, the Cl^−^ concentration on the treatments with 201 kg Cl/33 S showed a mean value for Cl^−^ of 98.5 mg·L^−1^ at 0–0.10 m depth and 66.4 mg·L^−1^ at 0.10–0.20 m depth, without significant differences on crop yield during the 4 years but with significant influence on N uptake and efficiency. These results indicate that for coffee, the ideal Cl^−^ concentration on the soil should be below 30 mg L^−1^. Regarding coffee, Brazil Santos et al. [21] found a significant difference in the Cl^−^ content in the soil, where the treatments with 100% K^+^ applied as MOP had an average Cl^−^ content in the soil in the first 20 cm of the depth of 130 kg·ha^−1^ (approx. 65 mg·L^−1^ of Cl^−^), while the treatment without Cl^−^ with 100% K^+^ applied as SOP had a significantly lower Cl^−^ content in the soil at the same depth equivalent to 50 kg·ha^−1^ (approx. 25 mg·L^−1^ of Cl^−^).

When Cl^−^ is applied to coffee at average doses of 100 to 150 kg·ha^−1^ in balance with S without exceeding foliar concentrations of 0.33% of DW and soil contents of 30 mg L^−1^, it has positive effects on the efficient use of nitrogen, increasing it from 9% to 14%. We can affirm that the effect of Cl^−^ on NUE is related to a reduction in compartmentalization and an increase in NO_3_^−^ translocation, but with no effect on NO_3_^−^ assimilation, due to the negative effect of high doses of Cl^−^ on nitrogen utilization efficiency (NU_T_E). High doses of Cl^−^ both in the greenhouse (>180 mg·L^−1^) and in the field (>150 kg·ha^−1^) reduce NUE due to a Cl^−^ saturation effect in the vacuole at the physiological level, but also due to Cl^−^ saturation at the soil level, generating competition with anion uptake, especially with NO_3_^−^. The positive effects of moderate doses of Cl^−^ on NUE may not be observable or reproducible under conditions where coffee plants grow with prolonged periods of water deficit, generating soil and tissue Cl^−^ concentrations above normal levels, as previously mention.

## 4. Materials and Methods

During the period of four years, two trials were carried out under greenhouse and field conditions, with the aim to test rates and ratios of Cl^−^ and S in coffee and their influences on growth, nutrient uptake, productivity, and NUE.

### 4.1. Greenhouse Trial

The greenhouse trial was located in Dülmen, Germany, at the Hanninghof Research Center of Yara International. The mean air temperature was 23.1 °C (±2.2 °C) with a maximal air temperature of 31.4 °C and minimal air temperature of 15.3 °C, mean relative humidity of 64% (±10%), and mean light intensity of 20.0 Klux during summertime. Supplemental light (300 mmol m^−2^ s^−1^ photosynthetic photon flux density) over a period of 12–14 h was given when natural light became insufficient.

Coffee seeds from the *Coffea arabica* var. Cenicafé 1 were pre-germinated in dark conditions with a mean temperature of 28 °C for 6 weeks using disinfected sphagnum-moos as a germination medium. Before the radicle emerged (BBCH scale 03-Arcila et al., 2002 [42], the pre-germinated seeds were moved to small containers with perlite as a growing medium. The seeds were allowed to germinate for 6 months. During this germination process, the plants received a nutrient solution once per week containing: N (7.6 mM), P (0.3 mM), K (1.7 mM), Mg (0.2 mM), Ca (0.9 mM), Fe (5.0 mM), Mn (2.9 mM), Zn (1.5 mM), Cu (0.6 mM), B (9.2 mM) and Mo (0.2 mM).

Once the plants reached three pairs of leaves that were completely open (BBCH scale 13; Arcila et al. [42]), they were transplanted in pots of 4.5 L. The aim of the greenhouse trials was to evaluate the influence of different Cl^−^ and S rates and proportions on growth and nutrient uptake and NUE. In this trial, several Cl^−^ and S rates and ratios were tested as follows: 0/200; 60/160; 120/120; 180/80; 240/40 and 300/0 Cl/S in mg.L^−1^. In this trial, the coffee plants were grown in coarse sand as a growing medium. All other nutrients were applied as a nutrient solution to the soil surfaces, without any foliar application. Pots were watered with a complete nutrient solution containing N (28.6 mM), P (1.5 mM), Mg (6 mM), Ca (4.5 mM), Fe (14.7 mM), Mn (8.4 mM), Zn (5.3 mM), B (5.2 mM), Cu (14.2 mM) and Mo (1.5 mM). The nutrient solution was applied once per week with application volumes between 60 and 120 mL, according to the water demand of the plants. The soil moisture was monitored daily with the aim of avoiding water deficit or excess, keeping it between 60% and 70% of the water holding capacity.

After 9 months of transplanting, the coffee plants were trimmed, and the tissues (leaves, stems, and branches) were dried in an oven at 65 °C until a constant weight was attained. The dried material was then finely ground for nutrient analysis in the lab.

Finely milled plant materials were used for elemental analysis after wet digestion in a microwave digester (MLS mega; MLS GmbH, Leutkirch, Germany). All the micro- and macronutrients (excluding nitrogen) were analyzed using inductively coupled plasma optical emission spectrometry (Perkin-Elmer Optima 3000 ICP-OES; Perking-Elmer Corp, Shelton, CT, USA). The nitrogen was determined by the micro-Kjeldahl method.

Two nitrogen use efficiency (NUE) indicators were calculated: (i) N use efficiency (NUE), calculated as the total nitrogen uptake by the shoot plant divided by total N applied during the growing period (g shoot DW g^−1^ N), and (ii) N utilization efficiency (NU_T_E), calculated as the total shoot dry biomass divided by the total nitrogen content in the shoot (g DW mg^−1^ N). The NU_T_E indicator allows for an understanding of how efficiently the transported N is used by the plant [8].

### 4.2. Field Trial

For four years, from January 2017 to December 2020, a field trial was carried out in the southeast region of Colombia, in Garzón-Huila on a farm located at 2°11′ N–75°34′ W at 1.437 m elevation. The soil predominantly comprised consolidated materials, was mainly granitic in nature, and had moderately low soil fertility, with characteristics as follows: pH 4,71; organic matter at 3.08%; and P, K, Ca, and Mg contents at 11, 125, 530, and 52 mg·kg^−1^, respectively; and soil particle distribution of 48% sand, 18% silt, and 34% clay. The pH was determined in water (1:1), organic matter by Walkley–Black, P by Bray-II, and the exchangeable fraction of K, Mg, and Ca with 1 N ammonium acetate extraction (1 N NH_4_C_2_H_3_O_2_, pH 7.0). The cations in the extracts were detected using an ICP (Perkin Elmer, Optima 8000, Shelton, CT, USA), and soil texture analysis was performed using the hygrometer Bouyoucos method. The mean climatic conditions observed in the region from 2017 to 2019 are shown in Table 4.

The trial was run using the *Coffea arabica* L. variety Castillo^®^ with resistance to the coffee leave rust (CLR) disease generated by the fungi *Hemileia vastatrix* Berkeley and Brome [42]. The plantation was established in 2012 in a full sunshine condition (FS) planted at a density of 6.666 plants·ha^−1^ at 1.0 m distance between plants and 1.5 m distance between rows. Before the treatment application, the plantation was stem trimmed at 30 cm height in January 2017, aiming to rejuvenate the plantation and to initiate a new productive cycle from 2018 to 2020. Five treatments differing in their Cl/S ratio were installed in the trial. Rates of Cl^−^ and S increased over the 4 years according to increments in growth and yield formation (Table 5).

The sources of Cl^−^ used in the trial were potassium muriate (MOP with 60% K_2_O and 46% Cl), and the S source was potassium sulfate (SOP with 50% K_2_O and 18% S). The Cl^−^ and S rates were adjusted based on the potassium requirement of the crop, considering crop K_2_O demand and K_2_O content in the soil [43,44,45,46]. The mean nutrient rates over 4 years were: 231 kg N, 265 kg K_2_O, 80 kg P_2_O_5_, 110 kg CaO, and 51 kg MgO kg·ha^−1^ (Table 5). The total nutrient rate was split into 3 applications during the year: at pre-flowering, and at 30 and 100 days after flowering. Fertilizer rates and application time are currently and commonly used by coffee farmers.

The experiment was set up in a randomized complete block design with four replications; each plot in the block had 45.0 m^2^ with 12 effective plants. Five plants of each compelling lot were selected for yield assessments, and the other 7 plants of the plot were chosen to make destructive sampling of the coffee cherries every 30 days after flowering until harvest with the aim to evaluate the nutrient concentration and nutrient uptake of the coffee cherries and beans over time. In each of the 7 plants, branch number 10, counted from the apex to the base, was labeled at the pre-flowering stage (BBCH scale 54–57), and the cherries of the labeled branch were harvested every 30 days from 30 to 240 days after flowering. Each effective plant served as a replicate.

The other 5 effective plants were used for the assessments at harvest. On average, the coffee was harvested over 8 months in a year with the main harvest (70% of the year) collected from September to November. The coffee cherries were harvested when they reached maturity at BBCH scale 88 at 240 days after flowering.

All cherry samples were dried at 60 °C for a few days until a constant weight was attained and finely ground for nutrient analysis. The nutrients K, Ca, Mg, S, and Cl were analyzed using inductively coupled plasma optical emission spectrometry (Perkin-Elmer 400; Perking-Elmer Corp., Norwalk, CT, USA). N was analyzed by Micro-Kjeldahl method.

The nitrogen use efficiency by the coffee beans (NUE) was estimated using an index between the total nitrogen uptake by the coffee beans at harvest (BBCH scale 88) divided by the total amount of nitrogen applied during the crop season. This NUE indicator is known as a fertilizer-based indicator [47] and can be considered similar to the partial N balance (PNB = plant N content per unit of fertilizer N applied) described by Doberman [48].

With the aim to evaluate the Cl^−^ concentration and distribution in the soil profile after three years of treatment application, soil samples were collected per treatment and replicated at a 25 cm lateral distance to the stem, where nearly 86% of the coffee roots are located [49]. Samples were taken at three soil depths (10, 30, and 50 cm).

A soil moisture index (SMI) was estimated using daily weather data. The SMI is defined as the ratio between actual volumetric soil moisture and volumetric saturation humidity [50], following the water balance approach described by Ramirez and Küsters [22].

All data were submitted to statistical analysis (ANOVA) according to the experimental design using the Statgraphics Centurion XV software package (Statgraphics Technologies, Inc.). The Shapiro–Wilk modified test was applied for normality testing and the residual vs. prediction test to evaluate the heterogeneity of variances. Fischer’s test was used to detect the treatments that significantly affected the ANOVA.

## 5. Conclusions

This work presents novel results regarding the influence of Cl^−^ application on coffee and its influence on growth, productivity, and NUE. After four years of research in the greenhouse and on the field, we can conclude that:-A balance between Cl^−^ and S is necessary for coffee with the aim to improve N uptake and NUE. For greenhouse conditions, there should be between 60 and 180 mg·L^−1^ of Cl^−^ and between 160 and 80 mg·L^−1^ of S. Plants without Cl^−^ or without S significantly reduce the NUE. At the field level, the balances should be between 100 and 150 kg Cl^−^ ha^−1^·year^−1^ and between 73 and 50 kg S·ha^−1^·year^−1^.-Cl^−^ is applied to coffee at rates as a “macronutrient”, and the coffee plants can take higher amounts of Cl^−^, reaching concentrations on the leaves of higher than 0.69% without toxicity symptoms, but with a significant reduction in dry biomass accumulation. With the aim to keep a balance between biomass accumulation and NUE, the Cl^−^ content in the leaves should be lower than 0.33%, placing the coffee into the group of glycophyte Cl^−^-sensitive plants.-Cl^−^ rates at the field level that are higher than 150 kg Cl^−^ ha^−1^·year^−1^ significantly increase the Cl^−^ content in the soil and Cl^−^ uptake by the coffee cherries and reduces N uptake by the coffee cherries, reducing the NUE on average by 9% to 14%.-In terms of the NUE, the Cl^−^ content in the soil in coffee in the first 20 cm depth should not exceed the concentration of 30 mg·L^−1^.-The NUE in coffee could be improved by using agronomical management practices such as the selection of mineral potassium fertilizer sources, with the main aim to reduce the Cl^−^ application rates and to limit them to no more than 100 to 150 kg Cl^−^ ha^−1^ with a fine balance with S rates, as mentioned before.

## Figures and Tables

**Figure 1 plants-12-02033-f001:**
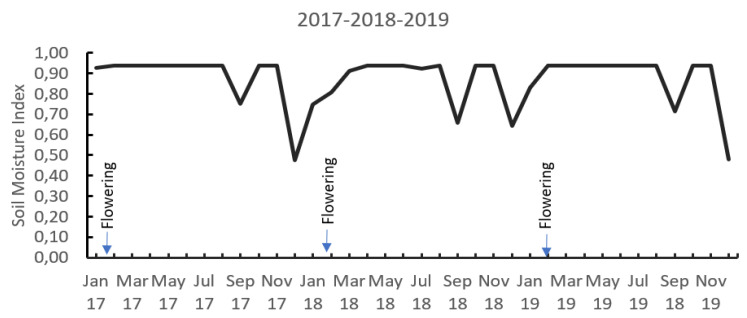
Soil moisture index distribution for 3 years (2017, 2018, and 2019). Soil moisture index means the fraction of the porous media of the soil that is occupied by water, SMI = 1.0 means that 100% of the soil porous is occupied by water, and SMI = 0 means that the soil is completely dry. (SMI was calculated seeing the methodology described by Ramirez et al. [22]).

**Figure 2 plants-12-02033-f002:**
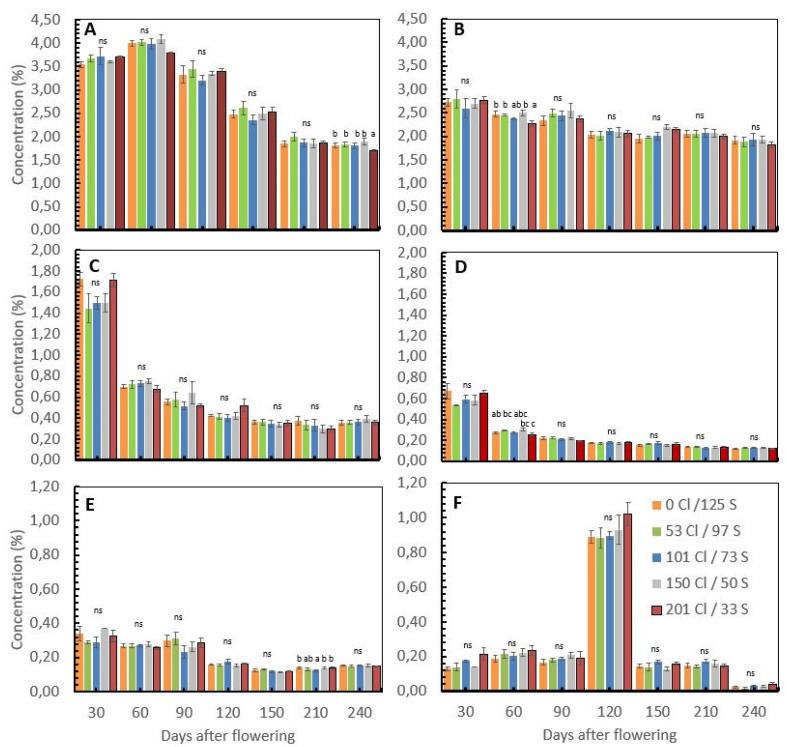
Influence of chloride/sulfur (Cl/S) rates on nutrient concentration in coffee cherries after flowering. Nitrogen (**A**), potassium (**B**), calcium (**C**), magnesium (**D**), sulfur (**E**), and chlorite (**F**). Values are the means of four replications ± standard error. Different letters denote statistically significant differences according to Fisher’s LSD test with *p* < 0.05, and ns denotes non-significant differences.

**Figure 3 plants-12-02033-f003:**
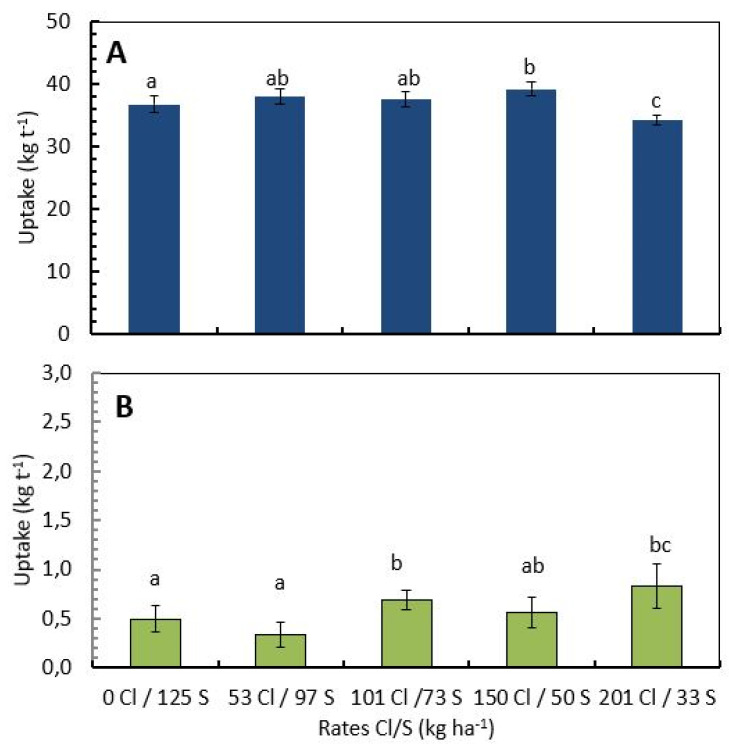
Influence of the chloride/sulfur rates on the nutrient demand per ton of green coffee beans. Nitrogen uptake (**A**), and chloride uptake (**B**). Values are the mean of four replications ± standard error; different letters denote statistically significant differences according to Fisher’s LSD test with *p* < 0.05.

**Figure 4 plants-12-02033-f004:**
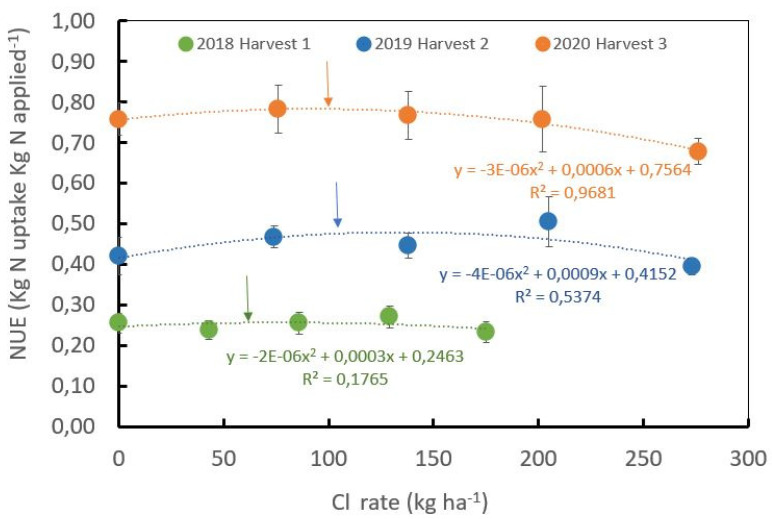
Influence of the chloride rates during three years of harvests on nitrogen use efficiency. Optimum Cl^−^ rates estimated from the first derivate of the function were 73, 113 and 110 kg·ha^−1^ for the harvest years of 2018, 2019 and 2020, respectively, ± standard error.

**Figure 5 plants-12-02033-f005:**
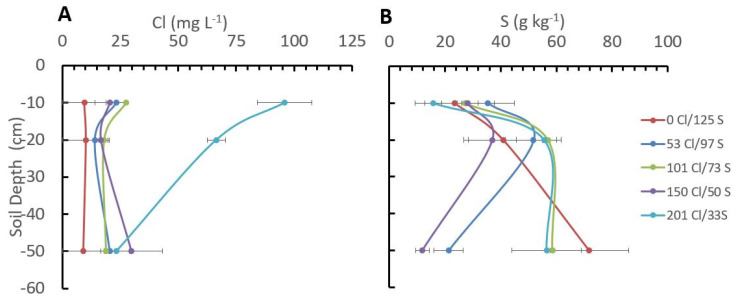
Influence of the chloride and sulfur rates on chloride (**A**) and sulfur (**B**) distribution in the soil profile. The bars in the figure represent the standard error.

**Table 1 plants-12-02033-t001:** Influence of different chloride (Cl^−^) and sulfur (S) rates on coffee plant biomass accumulation, nutrient uptake, and nitrogen use efficiency under greenhouse conditions.

Cl/S Rates	Total DW	Root DW	N	Cl	N Uptake	S Uptake	Total Shoot Cl Uptake	NU_T_E ^¶^	NUE ^¥^
mg L^−1^	g Plant^−1^		Content in Leaves %		mg Plant^−1^			g^2^ DW· mg^−1^ N	g DW·g^−1^ of N
0/200	82.1 d	27.0 c	2.58 a	0.03 a	1234.7 b	174.2 c	19.3 a	0.047 c	0.83 c
60/160	74.1 bc	20.3 bc	2.68 a	0.16 b	1230.0 b	137.2 c	89.1 b	0.044 c	0.86 cb
120/120	79.4 cd	23.3 bc	2.59 a	0.33 c	1236.5 b	106.0 c	173.4 c	0.046 c	0.86 cb
180/80	74.8 bc	18.5 b	2.69 a	0.55 d	1281.1 c	90.4 b	276.3 d	0.038 b	0.89 b
240/40	72.1 b	16.2 ab	2.67 a	0.69 e	1306.3 c	84.9 b	361.7 e	0.043 c	0.89 b
300/0	37.9 a	10.2 a	4.43 b	2.98 f	1095.8 a	19.4 a	587.2 f	0.029 a	0.73 a
*p* value	***	**	***	***	***	***	***	***	***

Different letters denote statistically significant differences according to Tukey’s test alfalfa = 0.05 ** *p* value < 0.05; *** *p* value < 0.01. ^¶^ Nitrogen utilization efficiency (NU_T_E total DW/total N content); ^¥^ Nitrogen use efficiency (NUE total N uptake by the shoot biomass/total N applied N).

**Table 2 plants-12-02033-t002:** Pr > F values from the statistical output (ANOVA) for the nutrient concentration after flowering and nutrient uptake at harvest time in the field trial.

Nutrient Concentration (%)								Nutrient Uptake at Harvest Time (t·ha^−1^)
Nutrient	Days After Flowering							
	30	60	90	120	150	210	240	240
	*p* Value							
N	0.67	0.13	0.48	0.37	0.61	-	0.099	0.028 **
K	0.90	0.02 **	0.64	0.93	0.03 **	0.94	0.83	0.239
Ca	0.13	0.50	0.46	0.25	0.91	0.42	0.82	0.220
Mg	0.38	0.04 **	0.52	0.85	0.49	0.83	0.19	0.398
S	0.38	0.68	0.18	0.36	0.60	0.081 *	0.54	0.459
Cl	0.14	0.69	0.78	0.53	0.11	0.21	0.27	0.006 **

* *p* value < 0.05; ** *p* value < 0.01.

**Table 3 plants-12-02033-t003:** Influence of tCl/S rates on coffee yield, nitrogen uptake, and use efficiency in coffee.

Cl/S Rate ^±^ kg·ha^−1^	Yield ^+^	N-Uptake per Ton	Total N^−^ Uptake per ha	Mean N. Applied per ha ^++^	NUE ^+++^
	t·ha^−1^	kg·t^−1^	kg·ha^−1^ year^−1^		kg·kg^−1^
0/125	4.02 ±2.05)	36.7 (±2.04)	147.4 (±74.5)	308 (±20.75)	0.48 (±0.23)
53/97	4.01 (±1.85)	39.1 (±2.43)	157.3 (±75.3)	308 (±20.75)	0.51 (±0.24)
101/73	4.03 (±1.96)	37.5 (±2.32)	151.4 (±75.9)	308 (±20.75)	0.49 (±0.23)
150/50	4.03 (±2.11)	37.9 (±2.08)	152.8 (±78.5)	308 (±20.75)	0.50 (±0.24)
201/33	3.92 (±1.87)	34.3 (±1.31)	134.3 (±64.21)	308 (±20.75)	0.44 (±0.20)
*p* value	ns	*	*		ns

^±^ The chlorate rate is the mean Cl^−^ applied during the whole season (2017–2020); ^+^ Mean yield for three years of harvest: 2018, 2019, 2020; ^++^ Mean nitrogen applied for four years (2017–2020); ^+++^ NUE is the ratio between N uptake by the coffee cherries/N applied during the season. NUE is total N uptake by the coffee cherries per ha/total N applied per ha; ns is not significant, and * is significant at *p* value < 0.10.

**Table 4 plants-12-02033-t004:** Climate conditions obtained from the weather station *.

Year	T·min (°C)	T·max (°C)	T. med (°C)	R.H (%)	Rainfall (mm)	Sunshine (h)	Rainfall in the Trial Area (mm)
2017	16.9	24.6	20.2	77.1	1.575	1.1467	2.483
2018	16.6 ^±^	25.1 ^±^	20.3 ^±^	72.6 ^±^	1.234	-	2.053
2019	16.1	23.8	19.4	-	1.426	1.314,0	2.319
2020	-	-	-	-	1.206	-	2.100
Mean	16.5	24.5	20.0	74.8	1.360	1.2303	2.238
1955–2010	16.2	24.5	19.7	-	1.330	1.2500	-

* Jorge Villamil weather station, 2°20′ N–75°31′ W, provided by the National Coffee Research Center—Meteorological Network. T. min, average minimum air temperature; T. max, average maximum air temperature; T. med, average mean air temperature; R.H., average relative humidity. ^±^ Mean value for three months.

**Table 5 plants-12-02033-t005:** Treatment description during the four-year trial.

Treatment	2017	2018	2019	2020	Average
	Cl/S Rates (kg·ha^−1^)				
1	0/41	0/96	0/174	0/188	0/125
2	21/30	43/62	74/140	74/156	53/97
3	40/20	86/41	138/109	138/123	101/73
4	61/10	129/22	205/78	202/91	150/50
5	80/0	175/30	276/44	276/59	201/33
1 to 5	K_2_O Rates (kg·ha^−1^)				
	110	230	360	360	265
	N Rates (kg·ha^−1^)				
	160	163	280	320	231
	P_2_O_5_ Rates (kg·ha^−1^)				
	90	46	92	92	80
	CaO Rates (kg·ha^−1^)				
	130	86	104	121	110
	MgO Rates (kg·ha^−1^)				
	29	50	54	72	51

## Data Availability

No applicable.

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
