# Peer review of "Influence of Variable Chloride/Sulfur Doses as Part of Potassium Fertilization on Nitrogen Use Efficiency by Coffee"

_plants, 2023, doi:10.3390/plants12102033_

Round 1

Reviewer 1 Report

The title should be change as Chloride/Sulphure ration Nutrition influences the Nitrogen Use efficiency in  Coffee.

If, in addition to the Cl/S ratio, soil pH and electrical conductivity are correlated, different approaches will emerge. It will not be enough to assert only the Cl/S ratio and the efficiency of using N.

Author Response

  • Title should be changed:

R/ Agree with the suggestion and done.

  • If a different approach will emerge in addition to the Cl/S ratio, soil pH and electrical conductivity are correlated. It will not be enough to assert only Cl/S ratio and the efficiency of using N.

R/

  • Strong correlation between pH and EC was observed in the soil during the field trial. The lowest EC (<0,4 dS.m-1) was observed in a pH >4,5.

  • Does not exist significant effects between Cl/S treatments and the soil pH and EC. Only exist significant effects between soil depths and pH or EC (See ANOVAs)
  • Critical level for coffee is higher than 1,2 dS.m-1.

  • Only at 20 cm of soil depth was found a correlation between EC and NUE (highest NUE at EC<0,4 dS.m-1). Interesting to observe that at 10 cm depth, the highest NUE is achieved at EC<1,0 dS.m-1. In coffee, significant dry biomass reduction is achieved when EC is >1,1 dS.m-1.(Sadeghian & Zapata, 2014)
  • At 20 and 50 cm depth positive correlation is observed between pH and NUE with highest NUE at pH >4,5 at 20 cm depth and 4,7 at 50 cm depth.

      The changes in the EC in the soil are more significantly correlated with the K content in the soil than with the Cl.

-    For coffee, the K content >0,4 Cmolc.Kg-1 is considered high, and in the field trial at that level, the EC is near of 0,7 dS.m-1, below the salinity threshold level.   

Reviewer 2 Report

Some figures are not clear enough, I recommend redrawing (Figure 2, Figure 4).

L32 I recommend changing " considered " to "considered as".

Author Response

  • Figures were redrawn
  • Paper will send after reviewers’ correction to English Editions.
  • Changes on L32 made.

Reviewer 3 Report

Dear colleagues!

Please find attached the reviews report

Author Response

  • C1: Title was changed integrating the recommendations of the reviewer 1 and 3.
  • Comments C2 to C8 changes made.
  • C9 “ L18-19: And what about the effect on the growth in field condition? You noted a significant effect?” – In the field trial, the objective was to evaluate the influence of the Cl/S rates on productivity, nutrient uptake by the coffee cherries, and NUE. Growing was not measured
  • C10: Conclusions in the abstract improved.
  • C11: done.
  • C12: done and references adjusted.
  • C13: The role of the Cl in the soil was included in the introductions, and new references were included (Lines 50 to 54 in the new version).
  • C14: a definition of the NUE on the cited references was described. In the Material and Methods session, a description of the NUE indicator used on both trials is described based on the indicators described in references 37 and 38.
  • The sentence’s authorship is from the author as a part of his knowledge of the coffee production systems, regions, coffee farmers, and the fertilizers consumption dynamic.
  • C16: This is a misunderstanding, bean yield makes references a green coffee beans yield. I changed to: “he economic optimum coffee yield is achieved with a mean rate of….”
  • C17: Paragraph adjusted.
  • C18: done.
  • C21: done.
  • done
  • C25: correction included
  • C27: In the case of the greenhouse trial all the p values on the analysis of variance are with at alfa <5%. A significance level of 1% was deleted.
  • C28: done.
  • C29: done
  • C30: done
  • C31: done
  • C32: done
  • C33: done
  • C34: done
  • C35: done, Discussion was improved and 10 new references were included.
  • C36: done
  • C37: done, the fertilizer rates and source were applied according to the farmers practices that follow the local research institutes.
  • C38: Mean climate condition were add from 1955 to 2010. (Table 4)

Round 2

Reviewer 2 Report

This manuscript investigates the chloride nutrition influences the nitrogen use efficiency in coffee. The work of this manuscript is beneficial to improve the nitrogen use efficiency in coffee.This version of the article is suitable for publication.

The language expression of this paper is clear.

Author Response

(The authors gave the same response as above.)

Reviewer 3 Report

The authors have made the necessary corrections. I have no further comments

Best Regards

Author Response

(The authors gave the same response as above.)
